# Reflectometry Reveals Accumulation of Surfactant Impurities at Bare Oil/Water Interfaces

**DOI:** 10.3390/molecules24224113

**Published:** 2019-11-14

**Authors:** Ernesto Scoppola, Samantha Micciulla, Lucas Kuhrts, Armando Maestro, Richard A. Campbell, Oleg V. Konovalov, Giovanna Fragneto, Emanuel Schneck

**Affiliations:** 1Biomaterials Department, Max Planck Institute of Colloids and Interfaces, Am Mühlenberg 1, 14476 Potsdam, Germany; micciulla@ill.fr (S.M.); lucas.kuhrts@mpikg.mpg.de (L.K.); 2Institut Laue-Langevin, 71 avenue des Martyrs, 38000 Grenoble, France; maestro@ill.fr (A.M.); richard.campbell@manchester.ac.uk (R.A.C.); fragneto@ill.fr (G.F.); 3Division of Pharmacy and Optometry, University of Manchester, Manchester M13 9PT, UK; 4ESRF, The European Synchrotron, 71 Avenue des Martyrs, 38000 Grenoble, France; konovalo@ill.fr; 5Institute of Condensed Matter Physics, Physics Department, TU Darmstadt, Hochschulstrasse 8, 64289 Darmstadt, Germany

**Keywords:** X-ray reflectometry, neutron reflectometry, fluorocarbons, liquid/liquid interfaces

## Abstract

Bare interfaces between water and hydrophobic media like air or oil are of fundamental scientific interest and of great relevance for numerous applications. A number of observations involving water/hydrophobic interfaces have, however, eluded a consensus mechanistic interpretation so far. Recent theoretical studies ascribe these phenomena to an interfacial accumulation of charged surfactant impurities in water. In the present work, we show that identifying surfactant accumulation with X-ray reflectometry (XRR) or neutron reflectometry (NR) is challenging under conventional contrast configurations because interfacial surfactant layers are then hardly visible. On the other hand, both XRR and NR become more sensitive to surfactant accumulation when a suitable scattering length contrast is generated by using fluorinated oil. With this approach, significant interfacial accumulation of surfactant impurities at the bare oil/water interface is observed in experiments involving standard cleaning procedures. These results suggest that surfactant impurities may be a limiting factor for the investigation of fundamental phenomena involving water/hydrophobic interfaces.

## 1. Introduction

Interfaces play key roles in wet-technological applications and are integral components of biological matter [1,2,3,4]. A large percentage of the mass of cells and tissues is organized in the form of molecular layers, such as biomembranes, whose surfaces are in contact with their aqueous surroundings and are essential for their functioning. In technology, the properties of interfaces determine, for instance, the stability of colloidal systems [2] and the transport of ions or molecules between two liquid phases [5]. Bare interfaces between water and hydrophobic media like air, oil, polymeric materials, or carbon-based materials are of special relevance for fundamental research and applications involving emulsions and foams [6,7,8]. Intensively studied aspects of hydrophobic interfaces include the adsorption of molecules [9,10], the preferential accumulation of ions [11,12], and water-mediated forces [13,14], among others.

Although many phenomena involving hydrophobic interfaces are well understood today, there are a number of observations which have eluded a consensus mechanistic interpretation: (1) Air and oil bubbles in water have an apparently negative zeta-potential [15,16] even though their surfaces are naturally considered uncharged. (2) Thin-film balance experiments on the interaction between charged solid surfaces and bare air/water interfaces revealed electrostatic repulsion requiring negative charges at the water surface [17,18,19]. (3) The surface tension of aqueous electrolyte solutions exhibits a characteristic minimum around a salt concentration of 1 mM irrespective of the preferential surface interaction of the ion type, known as the Jones–Ray effect [20,21].

Various mechanisms underlying such behavior have been invoked but remained subject of controversy [22,23]. More recently, a common explanation for all the above observations has been offered by Uematsu et al. based on an interfacial accumulation of charged surfactant impurities in water [24,25]. Experimental reports provoking [15,16,17,18,19,20,21] and further corroborating [26,27] this explanation, however, contain only indirect evidence, while experimental measurements directly quantifying the accumulation of surfactant impurities at bare water/hydrophobic interfaces have so far been missing. Ellipsometry is a very well-suited method to study molecular adsorption to liquid interfaces [28], but its accuracy relies on reference measurements involving truly pristine interfaces.

Unambiguous direct measurements of molecular excesses at soft interfaces can be obtained by X-ray and neutron scattering techniques, such as total-reflection X-ray fluorescence (TRXF) or reflectometry. Although TRXF allows quantification of the surface excess of chemical elements on an absolute scale [29], in order to quantify the interfacial accumulation of surfactants it would require rather detailed *a-priori* knowledge of their chemical composition and the presence of heavy-enough chemical elements (e.g., phosphorus or sulfur) contained exclusively in the surfactant molecules. In contrast, reflectometry with X-rays or neutrons [30,31] is sensitive to molecular layers in terms of their scattering length density (SLD) [32]. To this end, the SLD of hydrocarbon chains can be considered a feature common to all common surfactants. Reflectometry has been used routinely to quantify the interfacial adsorption of molecules to air/water interfaces [33] and more recently also to oil/water (O/W) interfaces [5,34,35,36,37,38,39,40,41]. But so far there have been no studies evidencing the accumulation of surfactant impurities at bare water/hydrophobic interfaces.

In the present work, we show that identifying such accumulation under conventional SLD contrast settings is challenging because surfactants are then only poorly visible. However, both X-ray reflectometry (XRR) and neutron reflectometry (NR) become more sensitive to surfactant accumulation when a suitable SLD contrast is generated by the use of fluorinated oil. With this approach, significant interfacial accumulation of surfactant impurities is observed in experiments involving standard cleaning procedures combined with water from standard sources of double-deionized water. Our results thus suggest that water purity in experiments may be a limiting factor for the investigation of fundamental phenomena involving water/hydrophobic interfaces.

## 2. Results and Discussion

### 2.1. Scattering Contrast of Hydrocarbon Chain Layers at Water/Hydrophobic Interfaces (Theory)

Scattering contrast arises from a difference between the SLDs of neighboring media. Reflectometry studies with a focus on the structure of bare interfaces between water and hydrophobic media are often carried out with two types of experimental configurations offering strong contrast between the two bulk media, as schematically illustrated in Figure 1A: (i) Hydrogenous or deuterated water (H${}_{2}$O or D${}_{2}$O, respectively) or mixtures thereof contacting air or gas [33,42,43], or (ii) H${}_{2}$O or D${}_{2}$O contacting hydrogenous hydrocarbon chains in the form of oil [5,34,35,36,37,38,39,40,41] or of oil-like molecules grafted to a solid surface [44]. Contrast-optimized NR studies dealing with hydrophobized solids [45,46,47] involved considerable structural complexity, making interpretation difficult in view of the limited $Q_{z}$-range [46]. Water/fluid interfaces are structurally simpler and therefore particularly well suited to give unambiguous results. NR studies on liquid/liquid interfaces involving thin water or oil films on planar solid substrates [48,49,50,51,52,53,54,55] usually used partially deuterated oils and/or H${}_{2}$O/D${}_{2}$O mixtures, were aimed at characterizing organic interfacial layers rather than the bare interfaces, and will be discussed further below.

The properties of surfactant hydrocarbon chains can be assumed to be very similar to those of hydrocarbon (HC) oils like dodecane (C${}_{12}$H${}_{26}$) in terms of packing density and chemical composition. The X-ray and neutron SLDs of a thin hydrocarbon chain layer belonging to interface-adsorbed surfactant impurities are therefore approximated as $\rho_{\mathrm{HC}}^{x}\approx \rho_{C12H26}^{x}$ = 7.32 × 10${}^{-6}$ Å${}^{-2}$ and $\rho_{\mathrm{HC}}^{n}\approx \rho_{C12H26}^{n}$ = −0.46 × 10${}^{-6}$ Å${}^{-2}$, respectively, see Table 2. The choice of C12 chains as representative of surfactant chains is somewhat arbitrary. It was made because this is a typical chain length of surfactants commonly used in laboratories for cleaning and because it is consistent with what is often assumed in the theoretical literature [24,25]. Importantly, the SLD of alkanes does not depend much on their length, such that essentially the same conclusions would be drawn if a different chain length had been chosen. The SLD value for neutrons, $\rho_{\mathrm{HC}}^{n}\approx$ −0.46 × 10${}^{-6}$ Å${}^{-2}$, is also relatively similar (Δρ≤ 0.5 × 10${}^{-6}$ Å${}^{-2}$) to those of vacuum and H${}_{2}$O, see Table 2. As a consequence, in the experimental configurations (i) and (ii), adsorbed surfactant impurities (when present) do not generate much SLD contrast with at least one of the two neighboring media. This point is illustrated in Figure 1B,C,E,F, where SLD profiles are exemplified for a 10–Å–thick layer of hydrocarbon chains adsorbed to the interface ($d_{\mathrm{HC}}$ = 10 Å, $\phi^{\mathrm{HC}}=1$, see Section 3). Panels B and E show theoretical interfacial profiles of the X-ray (B) and neutron (E) SLDs, respectively, at an air/water interface corresponding to configuration (i) in the absence (dashed lines) and in the presence (solid lines) of the adsorption layer. Panels C and F show the same for an oil/water interface corresponding to configuration (ii). Note that water was assumed to be D${}_{2}$O for the neutron SLD profiles (panels E and F). The interfacial roughness was arbitrarily assumed as $\sigma_{\mathrm{cw}}$ = 5 Å, which is comparable to or slightly lower than those reported for interfaces between water and oil [36,41], somewhat higher than that of air/water interfaces [42,56], and closely related to the interfacial tension γ according to capillary wave theory ($\sigma_{\mathrm{cw}}\propto \gamma^{-\frac{1}{2}}$) [36]. Intrinsic roughnesses were neglected for simplicity ($\sigma_{o}=\sigma_{w}=0$, see Section 3). In all cases, the hydrocarbon chain layer does not stand out from the SLD profile but merely affects the gradual transition between the SLDs of the two bulk media, due to the unfavorable contrast settings. As a consequence, the associated X-ray and neutron reflectivities presented in Figure 2 get merely diminished upon addition the hydrocarbon chain layer. The adsorption of the layer thus cannot be readily distinguished from a simple increase in the interfacial roughness $\sigma_{\mathrm{cw}}$ between the two bulk media as the effects are virtually the same.

In summary, it is seen that standard contrast settings for the investigation of bare water/hydrophobic interfaces render thin interfacial HC layers hardly visible, because they cannot be easily distinguished from at least one of the bulk media. This result suggests that numerous published reflectometry-based studies would not have identified surfactant impurity accumulation at bare water/hydrophobic interfaces even if significant accumulation had occurred.

In order to highlight the adsorption of surfactant impurities, optimized contrast settings are required. To this end, fluorinated oil such as perfluorooctane (PFO, C${}_{8}$F${}_{18}$) is ideally suited for a number of reasons. At first, it is prototypical of a hydrophobic medium, featuring an interfacial tension with water as high as γ = 51 mN/m [57] (experimental data obtained via profile analysis tensiometry are shown in the Appendix A), similar to that of hydrocarbon oils [35,58]. Secondly, fluorinated oil has outstanding transmittance to cold neutrons, which facilitates NR experiments [39,59]. Finally, and most importantly, its X-ray and neutron SLDs are both strongly different from those of hydrocarbon chains and water (see Table 2). The corresponding experimental configuration (iii) is schematically illustrated Figure 1A. Panels D and G of the same figure show the X-ray (C) and neutron (F) SLD profiles at the interface between water (D${}_{2}$O in case of neutrons) and fluorinated oil in the absence and presence of a 10–Å–thick HC layer. It is clearly seen that the layer stands out as a distinct minimum, offering considerable scattering contrast for both X-rays and neutrons. As a consequence, the associated normalized X-ray and neutron reflectivity curves (red lines) presented in Figure 2 strongly deviate from unity. Importantly, the deviation is in the positive direction, and the presence of the HC layer can thus be distinguished from roughness effects. Note that in the representation used in Figure 2, where the reflectivity in the presence of the adsorption layer, $R_{\mathrm{HC}}\left(Q_{z}\right)$, is divided by the reflectivity of the bare rough interface, $R_{\mathrm{bare}}\left(Q_{z}\right)$, the influence of the interfacial roughness is eliminated, so it can be inferred that this result is valid irrespective of the choice of $\sigma_{\mathrm{cw}}$. With that, we conclude that contrast settings involving fluorinated oil are well suited to detect the adsorption of surfactant impurities to water/hydrophobic interfaces.

### 2.2. Experiments on Interfaces between Water and Fluorinated Oil

In the following we present and discuss the results of X-ray and neutron reflectivity measurements on interfaces between water and PFO. As described in Section 3, water was either MilliQ water from a laboratory source, or D${}_{2}$O as received. In both cases the water was filled into the liquid cells with pre-cleaned glass bottles and reflectivity was measured after an equilibration time of at least 30 min. Collecting one reflectivity curve takes 30 min (XRR) or 3 h (NR), such that the adsorption kinetics could not be time-resolved. Figure 3A,B show X-ray reflectivity curves measured on the beamline ID10 of ESRF with two different samples, using MilliQ water. The dashed lines are theoretical reflectivity curves calculated for a bare, albeit rough interface between water and PFO (see Section 3). The simulated reflectivity curves clearly do not reproduce the experimental data points. Instead, they stay significantly below the experimental data even though the roughness parameter was allowed to reach the lower plausible limit for liquid interfaces with γ≈ 50 mN/m, $\sigma_{\mathrm{cw}}$ = 4 Å [36,41]. This observation indicates that the true SLD profile at the interface is more complex. Indeed, the experimental data can be reproduced (blue solid lines) when an interfacial minimum in the SLD profile with adjustable depth and extension is introduced into the model. In the first step, this SLD minimum is realized by inserting into the model an interfacial distribution of “voids”, $\Phi_{\mathrm{void}}\left(z\right)$, i.e., of a medium with zero SLD ($\rho_{\mathrm{void}}^{x}$ = $\rho_{\mathrm{void}}^{n}$ = 0, see Table 2), which displaces water and PFO from the interface as described in Section 3. In view of the limited *z*-resolution and in line with earlier reports [41,44,60], the reflectivity curves are mainly sensitive to the integral density deficit *D* at the interface, which for free voids conveniently coincides with the integrated void distribution, $D=D_{\mathrm{void}}$, where
(1)$$D_{\mathrm{void}}=\int \Phi_{\mathrm{void}}\left(z\right)dz.$$

The best-matching density deficits obtained in the two independent X-ray experiments using regular MilliQ water are *D* = 5.3 Å for sample 1 and *D* = 4.4 Å for sample 2, see also Table 1. As determined in a second set of reflectivity fits (see Appendix A), the same deficits would also result from interfacial HC distributions $\Phi_{\mathrm{HC}}\left(z\right)$ characterized by $D_{\mathrm{HC}}$ = 19.9 Å and $D_{\mathrm{HC}}$ = 13.8 Å, respectively (see Table 1), where
(2)$$D_{\mathrm{HC}}=\int \Phi_{\mathrm{HC}}\left(z\right)dz.$$

However, it would not be physically realistic to attribute the density deficits solely to the interfacial accumulation of molecules, because there are other effects reducing the density at the interface, notably the interfacial packing/configuration of solvent molecules [41,60]. Indeed, significant density deficits of typically D≈ 1 Å have been observed also under conditions where surfactant adsorption is essentially invisible [41,44,61,62,63]. Values of up to *D* = 2 Å were predicted by theoretical studies [64,65,66]. Slightly higher values were reported for contrast conditions where a contribution of surfactant adsorption cannot be entirely excluded [47,60]. Here, we consider *D* = 2 Å a reasonable upper estimate for the density deficit induced by the interfacial packing/configuration of solvent molecules, while the rest, which is still considerable, can be reasonably attributed to the adsorption of surfactant impurities.

To proceed along this line, the reflectivity data are re-interpreted with a more elaborate model, involving again a distribution of voids representing interfacial packing effects but with fixed integral $D_{\mathrm{void}}$ = 2 Å, and additionally an interfacial distribution of hydrocarbon chains. The HC distribution integrals obtained with this combination model are $D_{\mathrm{HC}}\approx$ 11 Å and $D_{\mathrm{HC}}\approx$ 5 Å (see Equation (Equation 2) and Table 1). These values must be considered a lower estimate of the interfacial HC accumulation required to reproduce the experimental data, because an upper estimate for $D_{\mathrm{void}}$ was used, whose persistence in presence of surfactants can strictly be assumed only in the limit of low surfactant adsorption. The corresponding simulated reflectivity curves are indicated with red solid lines superimposed to the blue solid lines in Figure 3A,B. The virtually exact overlapping of the two sets of lines reconfirms that the reflectivity data are primarily sensitive to the interfacial density deficit and not to its origin. Panels C and D show the associated SLD profiles featuring distinct interfacial minima. As seen in the profiles, the interfacial roughness is considerable. The presence of the layer however makes it difficult to disentangle contributions of the capillary roughness from roughness due to molecular protrusions and chemical heterogeneity. It should be noted that the fitting procedure requires that the HC distribution mainly overlaps with the aqueous hemi-space, in order to generate the density deficit and to reproduce the experimental data.

Coming back to the obtained distribution integrals, the values of $D_{\mathrm{HC}}$ can be understood as “equivalent thicknesses” [67] of the HC layers formed by adsorbed surfactants. Values of 5 to 11 Å accordingly correspond to a considerable accumulation of surfactants under the given experimental conditions. For sample 1 with $D_{\mathrm{HC}}\approx$ 11 Å, the accumulation is comparable to a full monolayer coverage of typical surfactants (e.g., with C${}_{12}$ chains), for sample 2 with $D_{\mathrm{HC}}\approx$ 5 Å, the accumulation is indicative of a partial coverage. The differences have to be attributed to slightly different surfactant impurity levels in the two samples. It should be noted, however, that the exact numbers depend on the choice of $\rho_{\mathrm{HC}}^{x}$, which reflects the packing density of the surfactants’ hydrocarbon chains at the interface. When a looser HC packing is assumed ($\rho_{\mathrm{HC}}^{x}<\rho_{C12H26}^{x}$), the same density deficit corresponds to lower values of $D_{\mathrm{HC}}$, i.e., lower surfactant coverage. But even if an extremely low density is assumed ($\rho_{\mathrm{HC}}^{x}=\rho_{\mathrm{methyl}}^{\times }$ = 4.56 × 10${}^{-6}$ Å${}^{-2}$ [68]), the coverage remains considerable, with $D_{\mathrm{HC}}\approx$ 5 Å and $D_{\mathrm{HC}}\approx$ 3 Å for samples 1 and 2, respectively.

We move on to the results obtained by NR. Figure 4 shows neutron reflectivity curves measured on the instrument FIGARO of ILL during two different beamtimes, with PFO contacting D${}_{2}$O. The dashed lines are simulated reflectivity curves assuming a rough interface between water and PFO without any additional layer. The roughness parameter was again allowed to go to the lower plausible limit of $\sigma_{\mathrm{cw}}$ = 4 Å for this type of interface (see above). The agreement between experimental and simulated reflectivities is good. In fact, the agreement becomes only marginally better (solid line) when the model accounts for an interfacial density deficit via implementation of a volume fraction profile of voids. The best-matching deficits are small for both samples, $D=D_{\mathrm{void}}\leq$ 2 Å (Table 1), equal within the confidence interval, and consistent with the reported deficits due to interfacial molecular structuring [41,44,60,61,62,63,64,65,66]. With that, no significant surfactant accumulation is detected in any of the two independent experiments. According to Figure 1G and Figure 2B the NR contrast is suitable for the detection of hydrogenous HC at the interface. This is illustrated in Figure 4B with a dotted line indicating the theoretical NR curve assuming the interfacial structure of sample 2 as obtained by XRR, including a HC distribution with $D_{\mathrm{HC}}\approx$ 5 Å. It is seen that the curve significantly deviates from the experimental data, however not to the extent suggested by Figure 2B, because the density deficit in the neutron SLD profile is considerably reduced when the HC distribution overlaps with the aqueous hemi-space (the medium with the higher SLD).

In summary, considerable surfactant adsorption to PFO/water interfaces is observed in the two XRR experiments (samples 1 and 2), while no such adsorption is observed in the two NR experiments (samples 3 and 4). One difference between the two sets of experiments is the use of regular MilliQ water from a laboratory source in the XRR experiments versus the use of D${}_{2}$O in the NR experiments. Organic contents of ≤5 ppb in MilliQ water (see Section 3) in principle do not exclude surfactant concentrations in the 100 nM range, sufficient to result in significant interfacial accumulation [24,25]. At these concentrations the overall surfactant amount in the aqueous volume (50 mL) is also roughly sufficient to result in surfactant (sub-)monolayers at the oil/water interface. We nevertheless clearly refrain from attributing the observed impurity adsorption to the water source. Instead, no matter whether MilliQ water or D${}_{2}$O is used, impurities may equally be introduced in the course of the sample preparation as a contamination of the used glassware or of the organic solvents used for cleaning. Irrespective of the contamination route, our reflectometry results strongly indicate that significant accumulation of surfactant impurities at bare water/hydrophobic interfaces can occur when standard cleaning procedures are used. The strongest evidence for the adsorption of impurities is that different samples exhibit very different values of the total density deficit *D*, which would be identical for all samples if it were only due to the intrinsic structure of the bare liquid/liquid interface. The extent of the surfactant accumulation, however, cannot be generalized because it will be highly dependent on the exact experimental procedures and the chemical nature of the impurities. Estimating the time scale for impurity adsorption to occur is also nontrivial, because uncontrolled convective material transport may dominate over diffusion-limited transport.

It is unlikely that the observed impurities are introduced through the PFO itself, because the only PFO-based impurities potentially detectable by XRR and NR would be under-fluorinated (i.e., partially hydrogenated) molecules of amphiphilic character, which are not among the common impurities in PFO [69]. Moreover, there is considerable variability between the different samples regarding the obtained density deficits *D* and adsorbed amounts $D_{\mathrm{HC}}$ even though PFO was introduced into the measurement cells from the same original bottle for samples 1, 2, and 4.

It should be noted that the highest accumulation levels observed here are much higher than the accumulation of charged surfactants estimated by Uematsu et al. [24,25] to be necessary to explain the phenomena mentioned in the introduction. However we emphasize that these theories consider *only* charged surfactants of one charge sign and do not account for the presence of uncharged or unlike-charged surfactants. In contrast, reflectometry is sensitive to the total adsorption of uncharged, positively charged, and negatively charged surfactants.

While fluorinated oils like PFO can be considered good generic models of hydrophobic surfaces, they are no chemically-accurate mimics of the more commonly occurring interfaces between water and hydrocarbon oils. To this end, fully or partially deuterated hydrocarbon oils (like C${}_{12}$H${}_{9}$D${}_{17}$) contacting D${}_{2}$O can be considered suitable for the detection of the interfacial adsorption of surfactant impurities by NR, as exemplified with a yellow line in Figure 2B. In fact, solid-supported thin films of partially deuterated hydrocarbon oils contacting D${}_{2}$O or H${}_{2}$O/D${}_{2}$O mixtures have been used to characterize molecular layers at oil/water interfaces [48,49,50,51,52,53,54]. Even though these studies were not aimed at identifying impurity adsorption to the bare interfaces, highly variable apparent roughness values reported in some cases for the bare reference surfaces [49,53] may be in principle influenced by surfactant impurity adsorption. Indication of surfactant impurity effects can also be found in the PhD thesis of one of the authors, where an adsorption layer was required to interpret NR data from interfaces between water and hydrocarbon oils [40].

## 3. Materials and Methods

### 3.1. Chemicals and Sample Preparation

Unless stated otherwise, organic solvents and heavy water (D${}_{2}$O) were purchased from Sigma (St. Louis, MO, USA) and used as received. Perfluorooctane (PFO, C${}_{8}$F${}_{18}$) was obtained from abcr GmbH (Karlsruhe, Germany). Perfluorocarbons like PFO have a high gas dissolving capacity, for which reason they are considered candidates as blood substitutes [70,71,72]. PFO was therefore de-gased by boiling it for 15 min before use. For XRR experiments, MilliQ water (H${}_{2}$O, MilliQ${}^{®}$ Integral ultrapure water Type 1, specific resistance ≥ 18.2 MΩ·cm, organic content ≤ 5 ppb) from a standard laboratory source was used. The liquid/liquid cells for XRR and NR [5,39,40] were cleaned by washing with organic solvents (chloroform, acetone, ethanol) and by plasma cleaning. Subsequently, the lower part of the liquid pool was rendered hydrophobic via covalent functionalization with octadecyltrichlorosilane (OTS) by exposure to a freshly-prepared solution of OTS in anhydrous toluene at a concentration of 1 mM for 30 min. After rinsing with toluene, the cells were dried in an oven at 70 ${}^{{}^{\circ}}$C for 1 h. PFO was poured into the cells directly from the bottle in which it had been delivered. Glassware for the transfer of water into the cells (bottles, beakers, and glass pipettes) was cleaned several times with organic solvents (chloroform, acetone, ethanol) and eventually rinsed with large amounts of MilliQ water directly from the source. Throughout the cleaning procedure gloves did not get into contact with any liquids reaching the interior of the cell or of the glassware used.

### 3.2. X-ray Reflectometry

Specular XRR experiments were carried out at the ID10 beamline of ESRF (Grenoble, France) [73] using double crystal deflector for the beam steering with a monochromatic beam with 22 keV photon energy (wavelength λ = 0.564 Å). This rather high energy is needed in order to reduce beam attenuation upon transmission through the condensed liquid media. The geometry was chosen such that the beam is transmitted through the supernatant aqueous phase of lower mass and electron density and is reflected upwards from the interface with the perfluorooctane phase of higher mass and electron density. The reflectivity *R*, i.e., the intensity of the reflected beam normalized by the intensity of the incident beam, was recorded as a function of the scattering vector component perpendicular to the interface,
(3)$$Q_{z}=\frac{4\pi }{\lambda }\sin\left(\theta_{i}\right),$$where $\theta_{i}$ is the incident angle. Measurements were carried out using an angle range of 0${}^{{}^{\circ}}$ < $\theta_{i}$ < 0.8${}^{{}^{\circ}}$ ($Q_{z}\leq$ 0.31 Å${}^{-1}$, see Equation (Equation 3)). The relative resolution, $\delta Q_{z}/Q_{z}$, was estimated as 1% during the data reduction procedure. Resolution was taken into account in the modeling process, by convolution with a Gaussian function of suitable width.

### 3.3. Neutron Reflectometry

Specular NR experiments were carried out on the FIGARO reflectometer [74] of Institut Laue-Langevin (ILL), where a polychromatic beam with a wavelength range 2 Å < λ < 20 Å was exploited. To access the range 0.005 Å${}^{-1}$ < $Q_{z}$ < 0.2 Å${}^{-1}$ (see Equation (Equation 3)), two incident angles, $\theta_{i}$ = 0.62${}^{{}^{\circ}}$ and $\theta_{i}$ = 2.70${}^{{}^{\circ}}$ were selected. The signal to noise ratio was improved by exploiting the unique reflection down feature of FIGARO. In this way the beam was transmitted through the heavier perfluorooctane phase, which exhibits much lower neutron beam attenuation than D${}_{2}$O [40,59]. Resolution of approximately 7% was taken into account by convolution with a Gaussian function, like for XRR.

### 3.4. Reflectivity Analysis

Reflectometry curves were analyzed with a fitting procedure based on parametrized volume fraction profiles of all chemical components [67]. Here, we account for water (“wat”), (fluorinated) oil (“oil”), surfactant hydrocarbon chains (“HC”), and additionally free voids (“void”) as pseudo-chemical-component. In the first step, the volume fraction profile of water at the bare interface with oil is described with an error function,
(4)$$\Phi_{0}^{\mathrm{wat}}\left(z\right)=\frac{1}{2}\cdot \left[1-\mathrm{erf}\left(\frac{z}{\sqrt{2}\sigma_{\mathrm{cw}}}\right)\right],$$where *z* denotes the distance to the interface and $\sigma_{\mathrm{cw}}$ the adjustable parameter characterizing the interfacial capillary wave roughness. Conversely, the volume fraction profile of oil can be described as
(5)$$\Phi_{0}^{\mathrm{oil}}\left(z\right)=1-\Phi_{0}^{\mathrm{wat}}\left(z\right).$$

The volume fraction profile of adsorbed species, initially neglecting the capillary wave roughness, is described as a rough layer with adjustable thickness *d* and adjustable volume fraction $\psi^{\mathrm{layer}}$ given by the difference of two error functions,
(6)$$\Phi_{\mathrm{layer}}^{\mathrm{flat}}\left(z\right)=\frac{1}{2}\psi^{\mathrm{layer}}\cdot \left[\mathrm{erf}\left(\frac{z-z_{0}+d/2}{\sqrt{2}\sigma_{o}}\right)-\mathrm{erf}\left(\frac{z-z_{0}-d/2}{\sqrt{2}\sigma_{w}}\right)\right]$$where $z_{0}$ represents an adjustable offset of the layer’s center of mass with respect the O/W interface. The parameters $\sigma_{o}$ and $\sigma_{w}$ represent the intrinsic roughnesses of the layer towards oil and water in terms of molecular protrusions and chemical heterogeneity. For simplicity, such an intrinsic roughness is neglected for the chemically very sharp bare oil/water interface (see Equation (Equation 4)).

In order to account for the influence of the capillary wave roughness on the layer profile, $\Phi_{\mathrm{layer}}^{\mathrm{flat}}\left(z\right)$ is subsequently convoluted with a Gaussian function of width $\sigma_{\mathrm{cw}}$:(7)$$\Phi_{\mathrm{layer}}\left(z\right)=\frac{1}{\sqrt{2\pi }\sigma_{\mathrm{cw}}}\int_{-\infty }^{+\infty }\Phi_{\mathrm{layer}}^{\mathrm{flat}}\left(z\right)\cdot e^{-\frac{{\left(z-\zeta \right)}^{2}}{2\sigma_{\mathrm{cw}}^{2}}}d\zeta$$

In the next step, Equations (Equation 6) and (Equation 7) are used to describe the volume fraction profiles of either HC, voids, or a combination of the two by assuming
(8)$$\psi^{\mathrm{layer}}=\psi^{\mathrm{HC}}+\psi^{\mathrm{void}}.$$

From that, we obtain
(9)$$\begin{array}{cc} \Phi_{\mathrm{HC}}\left(z\right) & =\Phi_{\mathrm{layer}}\left(z\right)\cdot \frac{\psi^{\mathrm{HC}}}{\psi^{\mathrm{layer}}} \end{array}$$
(10)$$\begin{array}{cc} \Phi_{\mathrm{void}}\left(z\right) & =\Phi_{\mathrm{layer}}\left(z\right)\cdot \frac{\psi^{\mathrm{void}}}{\psi^{\mathrm{layer}}}. \end{array}$$

Accounting only for voids ($\psi^{\mathrm{HC}}=0$) yields $\Phi_{\mathrm{void}}\left(z\right)=\Phi_{\mathrm{layer}}\left(z\right)$ and, in analogy, accounting only for HC ($\psi^{\mathrm{void}}=0$) yields $\Phi_{\mathrm{HC}}\left(z\right)=\Phi_{\mathrm{layer}}\left(z\right)$. The volume not occupied by HC or voids,
(11)$$\Phi_{\mathrm{unocc}}\left(z\right)=1-\Phi_{\mathrm{layer}}\left(z\right),$$is filled by water and oil according to
(12)$$\Phi_{\mathrm{wat}}\left(z\right)=\Phi_{0}^{\mathrm{wat}}\left(z\right)\cdot \Phi_{\mathrm{unocc}}\left(z\right)$$and(13)$$\Phi_{\mathrm{oil}}\left(z\right)=\Phi_{0}^{\mathrm{oil}}\left(z\right)\cdot \Phi_{\mathrm{unocc}}\left(z\right).$$

With all volume fraction profiles at hand, the corresponding X-ray and neutron SLD profiles, $\rho^{x}\left(z\right)$ and $\rho^{n}\left(z\right)$ are then calculated as
(14)$$\rho^{\left(x,n\right)}\left(z\right)=\rho_{\mathrm{wat}}^{\left(x,n\right)}\cdot \Phi_{\mathrm{wat}}\left(z\right)+\rho_{\mathrm{oil}}^{\left(x,n\right)}\Phi_{\mathrm{oil}}\left(z\right)+\rho_{\mathrm{HC}}^{\left(x,n\right)}\cdot \Phi_{\mathrm{HC}}\left(z\right)+\rho_{\mathrm{void}}^{\left(x,n\right)}\cdot \Phi_{\mathrm{voids}}\left(z\right),$$where $\rho_{\mathrm{wat}}^{\left(x,n\right)}$, $\rho_{\mathrm{oil}}^{\left(x,n\right)}$, $\rho_{\mathrm{HC}}^{\left(x,n\right)}$, and $\rho_{\mathrm{void}}^{\left(x,n\right)}$ are the X-ray and neutron SLDs of water, oil, HC, and voids respectively. Because $\rho_{\mathrm{void}}^{x}=\rho_{\mathrm{void}}^{n}=0$, the last term in Equation (Equation 14) trivially drops out. All other SLDs are calculated as [75]
(15)$$\rho^{\left(x,n\right)}=\frac{1}{v_{m}}\sum_{k}N_{k}\varrho_{k}^{\left(x,n\right)}$$where the index *k* identifies the type of nucleus, $N_{k}$ its number per molecular volume $v_{m}$ and $\varrho_{k}^{\left(x,n\right)}$ its complex X-ray or neutron scattering length [59]. The SLDs of the components used in the present work are summarized in Table 2.

Theoretical reflectivity curves are calculated by discretizing the SLD profiles into 1–Å–thick layers of constant SLD and subsequent application of Parratt’s recursive procedure [76]. In fits to experimental data the best-matching model parameters are obtained by minimization of the total squared difference $\chi^{2}$ between all theoretical and experimental values. Sampling of the physically-plausible parameter space is performed with an in-house routine based on the Metropolis Monte Carlo algorithm [77] with random variation of parameters. This approach has been validated by previous studies [5,39,78] and proven to be consistent with commonly used software [79]. Confidence intervals represent one standard deviation of the probability distributions $p\left(\Lambda \right)=\exp\left[-\left(\chi^{2}\left(\Lambda \right)-\chi_{\min}^{2}\right)/\chi_{\min}^{2}\right]$ associated to each parameter set Λ. Note that this procedure is self-consistent but not unique [67].

## 4. Conclusions

By using X-ray and neutron reflectometry with suitable contrast settings we have outlined how significant accumulation of surfactant impurities at bare oil/water can be examined. In the former case, the phenomenon was observed when regular laboratory sources of double-deionized water are used in combination with standard cleaning procedures. Although the source of the contamination in the experiments that were conducted has not been resolved, the results suggest that extremely high purity standards are required in future studies in order to prevent accumulation of surfactants contaminating water/hydrophobic interfaces which are supposed to be “bare”, as has been already suggested in early studies in the field [56]. Such contamination threatens to impede the experimental investigation of fundamental phenomena at bare water/hydrophobic interfaces such as ion-specific preferential interactions, water-structuring, or hydrophobic forces, among others. Therefore this issue merits more investigation to resolve how general our observations may apply to these phenomena. To this end, valuable insight may be gained by the in-situ combination of reflectometry with tensiometry or zeta-potential measurements. In summary, our findings lend credibility to the ideas of Uematsu et al. [24,25] and corroborate the general notion that bare hydrophobic interfaces in practice do not exist very long when in contact with water unless special measures are taken.

## Figures and Tables

**Figure 1 molecules-24-04113-f001:**
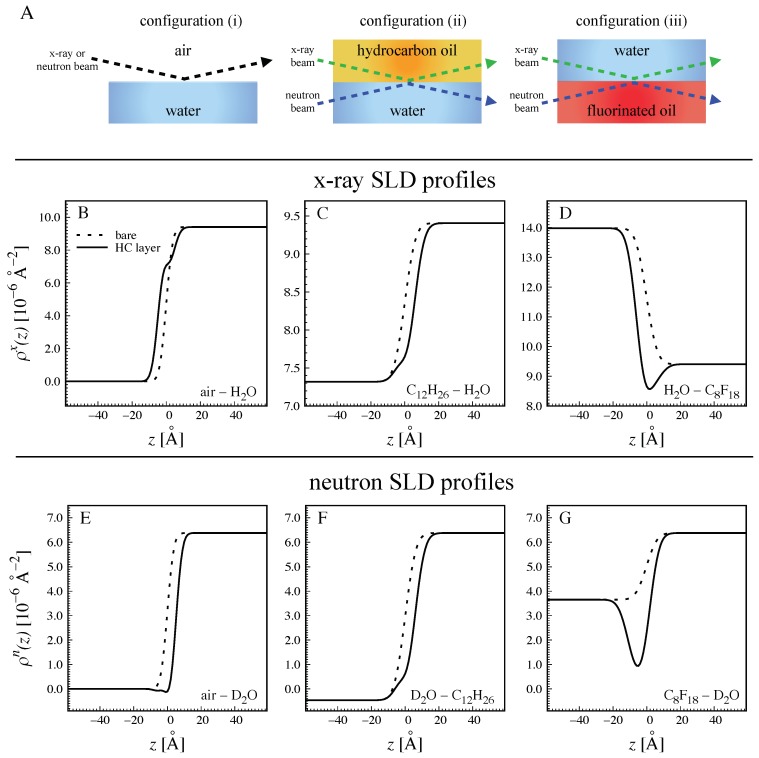
Experimental configurations for reflectometry on water/hydrophobic interfaces. (**A**) Schematic illustrations of (i) water contacting air, (ii) water contacting hydrogenous hydrocarbon oil, and (iii) water contacting fluorinated oil. Arrows indicate the paths of the X-ray or neutron beams, respectively. (**B**–**D**) Theoretical X-ray SLD profiles $\rho^{x}\left(z\right)$ based on estimated parameters for configurations (i–iii) with (solid lines) and without (dotted lines) a 10–Å–thick hydrocarbon (HC) layer adsorbed to the interface. (**E**–**G**) Theoretical neutron SLD profiles $\rho^{n}\left(z\right)$ for configurations (i–iii) with (solid lines) and without (dotted lines) a 10–Å–thick hydrocarbon (HC) layer adsorbed to the interface. The full set of corresponding reflectivity curves is shown in the Appendix A.

**Figure 2 molecules-24-04113-f002:**
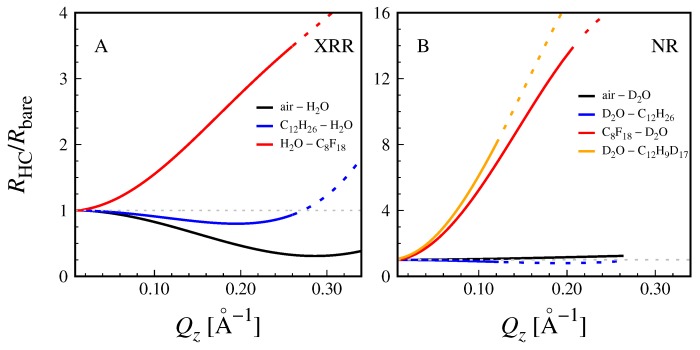
Theoretical reflectivity curves with a 10–Å–thick interfacial hydrocarbon (HC) layer, normalized by the theoretical reflectivity curves from the bare water/hydrophobic interfaces for various combinations of bulk media according to Figure 1 for XRR (**A**) and NR (**B**). The typically accessible $Q_{z}$-range before reaching the background is indicated with a solid line style. The dotted horizontal line indicates unity, i.e., no influence of the HC layer on the reflectivity curves.

**Figure 3 molecules-24-04113-f003:**
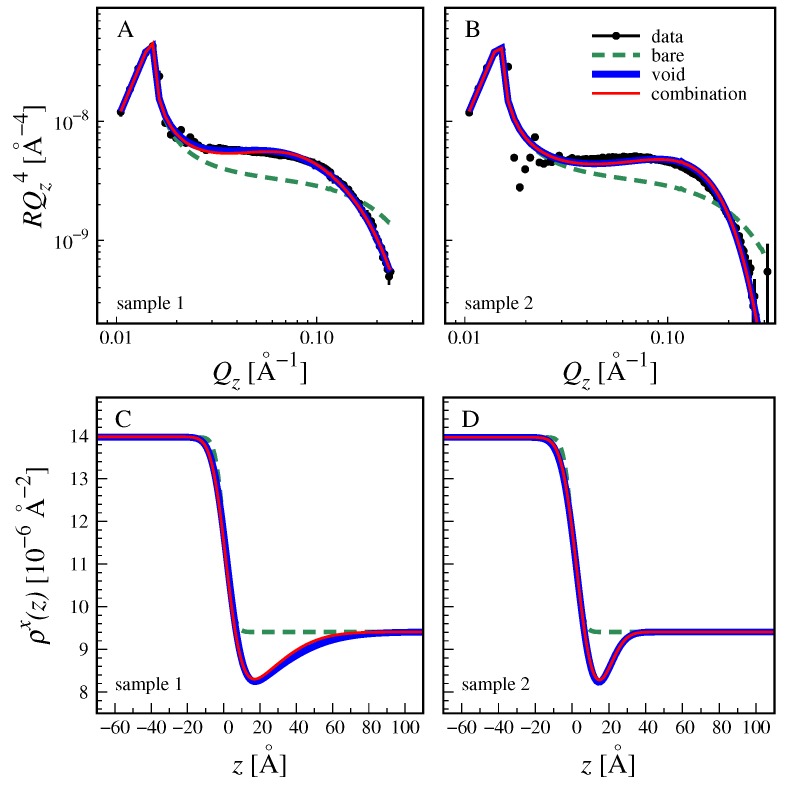
(**A**,**B**) Experimental X-ray reflectivity curves (symbols) of interfaces between water and PFO measured for sample 1 (**A**) and sample 2 (**B**). Dashed lines: theoretical reflectivity curves of the bare interface. Solid lines: theoretical reflectivity curves accounting for a distinct interfacial density deficit in the form of free voids (blue line) or due to a combination of free voids and hydrocarbon chains (red line). (**C**,**D**) Associated interfacial profiles of the X-ray SLD.

**Figure 4 molecules-24-04113-f004:**
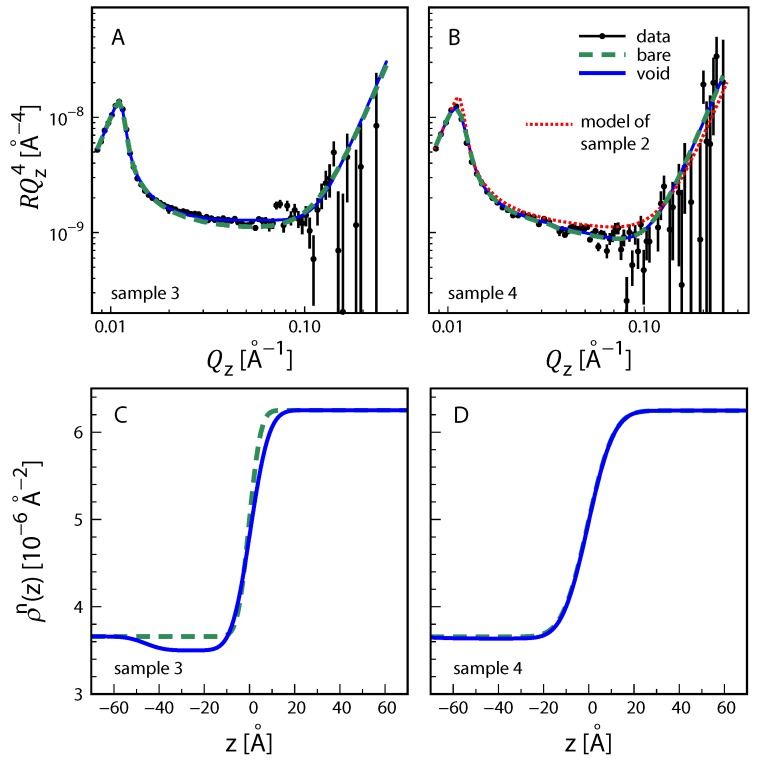
(**A**,**B**) Experimental neutron reflectivity curves (symbols) of interfaces between water and PFO measured for sample 3 (**A**) and sample 4 (**B**). Dashed lines: theoretical reflectivity curves of the bare interface. Solid lines: theoretical reflectivity curves accounting for a distinct interfacial density deficit in the form of free voids. Dotted line in panel B: theoretical reflectivity curve assuming impurity adsorption as determined by XRR for sample 2. (**C**,**D**) Associated interfacial profiles of the neutron SLD.

**Table 1 molecules-24-04113-t001:** Equivalent thicknesses in terms of integrated volume fraction profiles of voids ($D_{\mathrm{void}}$, first row), hydrocarbon chains ($D_{\mathrm{HC}}$, second row), and their combination (third row), as obtained in the analysis of experimental X-ray and neutron reflectivity data. Subscript and superscript values indicate the distances to the lower and upper limits, respectively, of the confidence intervals.

	XRR		NR	
	Sample 1	Sample 2	Sample 3	Sample 4
$D_{\mathrm{void}}$ [Å]	5.3${|}_{-1.2}^{+0.6}$	4.4${|}_{-3.2}^{+1.4}$	1.9${|}_{-1.8}^{+1.9}$	0.4${|}_{-0.3}^{+0.6}$
$D_{\mathrm{HC}}$ [Å] $\left(D_{\mathrm{void}}=0\right)$	19.9${|}_{-3.8}^{+2.8}$	13.8${|}_{-2.7}^{+5.0}$	-	-
$D_{\mathrm{HC}}$ [Å] $(D_{\mathrm{void}}$ = 2 Å)	11.3${|}_{-1.1}^{+2.9}$	5.5${|}_{-3.8}^{+3.2}$	-	-

**Table 2 molecules-24-04113-t002:** X-ray and neutron SLDs, $\rho^{x}$ and $\rho^{n}$, respectively, and molecular volumes $v_{m}$ calculated for all chemicals considered in this work. The molecular volume of C${}_{8}$F${}_{18}$ (perfluorooctane, PFO) was calculated by considering the molar weight and density reported by the supplier.

	$\rho^{x}$	$\rho^{n}$	$v_{m}$
	[$10^{-6}$ Å${}^{-2}$]	[$10^{-6}$ Å${}^{-2}$]	[Å${}^{3}$]
H${}_{2}$O	9.41–3.4 × 10${}^{-3}i$	−0.56	29.98
D${}_{2}$O	9.39–3.4 × 10${}^{-3}i$	6.37	30.04
C${}_{12}$H${}_{26}$	7.32–8.0 × 10${}^{-4}i$	−0.46	377.28
C${}_{8}$F${}_{18}$ (PFO)	13.92–7.8 × 10${}^{-3}i$	3.64	425.68
void	0.0	0.0	-

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
