# Peer review of "Reflectometry Reveals Accumulation of Surfactant Impurities at Bare Oil/Water Interfaces"

_molecules, 2019, doi:10.3390/molecules24224113_

Round 1

Reviewer 1 Report

Manuscript has clearly stated goal, well described and adequate methods. It points to some "hidden" aspects of the reflectometric studies on the surfactant-affected water-unpolar interfaces. It may be important for those who studied or will studied such interfaces by reflectometric techniques. Findings in these manuscript help to adequate and "impurity-free" evaluation of data on these systems. I have no suggestions to changes and in my opinion this manuscript can be accepted as it stands.

Reviewer 2 Report

The manuscript describes x-ray and neutron reflectometry study on the “bare” oil/water interface.  This work seems originally motivated to characterize the “void” layer at the oil/water interface.  The void layer was unambiguously determined at the solid-supported hydrophobic interface of water in the past.  The authors’ idea is to use fluorinated oil instead of normal oil to enhance the sensitivity of both reflectometry methods to the void layer.  What the authors found is how sensitive both methods (especially x-ray reflectometry) are to surfactant impurities when they used fluorinated oil.  Their idea to use fluorinated oil and their findings are both very interesting and the manuscript is worth to be published in Molecules.  However, I have some concerns described below and therefore recommend major amendments before publication.

The authors care about impurities originally existing in water and entering during experimental procedures.  How about impurities in perfluorooctane?  I checked the supplier website and found 99% purity for the material.  What is the rest 1%?  An answer would be non or partly fluorinated octane.  Don’t the 1% impurities accumulate at the interface?  I am not sure which of perfluorooctane and octane is more surface-active at the oil/water interface but am sure that the authors can discuss this point in the main text with some reference papers.  By the way, perfluorooctane has a lower boiling point than octane, so the boiling de-gassing technique may increase impurity fraction if reflux is not used.  Also what if other chemicals such as catalysis for fluorination reaction are inside the perfluorooctane as impurities?  Such chemicals may have low x-ray SLD and relatively high neutron SLD (to cancel void neutron SLD), which can completely explain the results in Figs 3 and 4.  In summary, the authors should describe the followings:

purity of perfluorooctane (99%) whether octane, the most plausible impurity, is accumulated at this perfluorooctane/water interface or not that other impurities in perfluorooctane may be able to explain Figs. 3 and 4.

Also, if XRR and NR data on perfluorooctane/air interface and the analysis results are available, they would be worthful to be shown in SI.

Miscellaneous and typos

(p.5, Fig.3A,3B)

Experimental data points are difficult to see. Thinner fitting lines or larger data points would be better.

(p.8,l.208)

C12H9O17 -> C12H9D17

(p.8,l.216)

war -> was

(p.9,l.225)

M Ohm m -> M Ohm cm

(p.10, eqs 4-7)

In eqs 4 and 5, they implicitly assumed zero sigma_w and zero sigma_o.  In contrast, eqs 6 and 7 shows that the parameters can be nonzero and different.  Can they solve this self-inconsistency?

(p.10, two lines lower from eq 5)

Phi^layer -> phi^layer

(p.10, eq 6)

Should the two erfs be the other way around?  Phi becomes negative.

(p.11, l.261)

Is chi_min^2 in the denominator necessary?  chi^2 is normalized per se.

Reviewer 3 Report

The article concerns the possible presence of surface-active impurities in water, which are very tough to detect but still can strongly influence multiple practical applications. This question is definitely very important for the correct interpretation of various experimental results.

The article is globally well written and can be interest for wide public. Still I have some questions.

Maybe, the most important concern: how authors can be sure that the fluorinated oil cannot be a source of surface-active molecules. Of course, the solubility of organics is very limited in fluorinated oils, but we are discussing about tiny concentrations of impurities.

Why do authors consider that the dodecane molecules can approximate the impurities? Could shorter (like ethanol?) or longer (proteins?) hydrocarbon chains better represent the impurities? What is the source of these impurities?

What is the difference between the samples 1 and 2; 3 and 4?

The kinetics of surface-active impurities absorption should change the NR and XRR signal with time. Is it true?

Surface tension/surface elasticity are very sensitive to the presence of impurities. I wonder if authors have measured, for example, the surface tension vs time dependence. This could be very helpful.

Is it possible to use a better cleaning procedure and more purified water to show that the XRR signal approaches the bare interface limit?

Round 2

Reviewer 2 Report

The authors revised the manuscript adequately according to the reviewers' comments.  Now I am happy to recommend the publication.